# Gastrointestinal Mast Cell Tumor in an African Dormouse (*Graphiurus* sp.)

**DOI:** 10.3390/vetsci9090497

**Published:** 2022-09-11

**Authors:** Yen-Chi Chang, Jung-Chin Chang, Jo-Wen Chen, Ying-Chen Wu, Ter-Hsin Chen

**Affiliations:** 1Graduate Institute of Veterinary Pathobiology, National Chung Hsing University, Taichung 40227, Taiwan; 2Jong-Shing Animal Hospital, Kaohsiung 80452, Taiwan

**Keywords:** CD117, *Graphiurus* sp., histopathology, immunohistopathology, KIT, mast cell tumor, Rodentia, toluidine blue stain

## Abstract

**Simple Summary:**

The African dormice, also known as micro squirrels, are now becoming more and more popular as they present as laboratory animals, exhibited animals, or pets. However, we know very little about the diseases in these tiny rodents, such as tumors or infectious microorganisms. A 3-year-old, male African dormouse was taken to the veterinary clinic after showing vomiting and anorexia. A stomach lesion was noted through imaging examinations. The patient died after being treated for 2 months. The veterinary pathologist diagnosed the lesion as a gastric mast cell tumor, cancer occurs mostly in cats and rarely in other animals. This is the first time that a mast cell tumor is diagnosed in an African dormouse. We suggest that the mast cell tumor in digestive tracts should be listed in the differential diagnoses when an elder African dormouse or other small rodent shows chronic digestive signs.

**Abstract:**

Mast cell tumors (MCTs) are well-known neoplasms derived from either mucosal or connective tissue mast cells. While well studied in several domestic species, MCTs are rarely documented in rodents. A three-year-old, male African dormouse (*Graphiurus* sp.) presented with a history of vomiting and anorexia for 3 months. Sonography revealed thickened gastric mucosa and hyperperistalsis. The patient died after receiving symptomatic treatment for 2 months. At necropsy, locally extensive, pale, thickened mucosal foci obscuring the first half of the stomach lumen was noted. Histological examination revealed moderately polymorphic, round, oval to spindle cells with amphophilic cytoplasmic granules infiltrating the mucosa to tunica muscularis, with moderate numbers of eosinophils. The mucosa was severely ulcerated with the proliferation of granulation tissue. The granules in most tumor cells exhibited metachromasia with the toluidine blue stain. Neoplastic cells revealed positive membranous immunoreactivity to KIT. Herein, we report the first case report of MCT in dormouse but also the first gastrointestinal MCT in a rodent species.

## 1. Introduction

Mast cells are derived from CD34+ precursor cells and are distributed throughout the body in connective tissues adjacent to small blood and lymphatic vessels [1,2]. When facing antigens or other foreign substances, activated mast cells release various cytokines and chemokines into the blood (degranulation) [1,2]. Therefore, mast cells contribute significantly to the initiation of immune responses and allergic reactions [1,2].

Mast cell tumors (MCTs) are very common in domestic animal species such as dogs, cats, and cattle [3]. The clinical presentation, anatomic location, and biological behavior of MCTs vary across species [2,4]. MCTs of the gastrointestinal tract (GIMCTs) are most frequently identified in cats and are occasionally reported in other domestic species [5,6]. In regard to rodents, spontaneous MCTs are relatively rare and are largely reported in laboratory species [7,8]. Cutaneous MCTs were reported in hamsters and a Richardson’s ground squirrel *(Spermophilus richardsonii*) [9,10]. No GIMCT has been reported in Rodentia.

African dormice (*Graphiurus* spp.) are mouse-sized, arboreal, nocturnal rodents that belong to the Gliridae family [11,12,13]. In the wild, dormice are group living, widely distributed throughout Africa, and are classified as Least Concern by the International Union for Conservation of Nature [13]. In recent years, African dormice are now an established captive species for laboratory animals, zoo collections, and companion animals [13,14]. Several studies involving captive *Graphiurus* spp. provide detail on behavior, feeding, and husbandry [13,14,15]. Nevertheless, information on diseases in African dormouse is rare and most studies are focused on the zoonotic aspect. In 2003, a shipment of African rodents including *Graphiurus* spp. introduced the monkeypox virus into the U.S. [16]. Antibodies against *Yersinia pestis*, the causative agent of bubonic plague, were detected in one African dormouse in a serological survey in northern Tanzania [17]. However, no neoplasm has been documented in *Graphiurus* spp.

Here, we present the case of gastric MCT in an African dormouse with gastric ulceration. To our knowledge, this is the first report of MCT in dormouse but also the first GIMCT in a rodent species.

## 2. Case Presentation

A three-year-old, intact, male African dormouse (*Graphiurus* sp.) was referred to the Jong-Shing Animal Hospital for a history of vomiting and anorexia continuing for 3 months. At initial presentation, physiological examination revealed no significant finding except retching. Symptomatic medication including Trimethoprim-Sulfamethoxazole (20 mg/kg BW), metoclopramide (1 mg/kg BW), mosapride (0.5 mg/kg BW), famotidine (1 mg/kg BW) was given twice a day for 1 week. At the follow-up visit, the owner claimed that the clinical signs did not resolve. The patient vomited transparent mucous fluid in the clinic. Cytological examination of the fluid revealed moderate number of neutrophils, eosinophils, and mucosal epithelium clusters. Occasionally, there were 10–15 μm in diameter, oval cells with amphophilic cytoplasm and round, euchromatic nuclei. Whole-body radiography revealed distended intestines and increased opacity of the lungs (Appendix A). Abdominal sonography revealed hypoechoic, noncircumferential, thickened gastric mucosa (Figure 1) and hyperperistalsis of the intestines. Minocycline (5 mg/kg BW), cetirizine (1 mg/kg BW), and maropitant (1 mg/kg BW) were added in the oral medication. After treatment for one month, the patient did not show any improvement. The patient was found dead 16 days later.

At necropsy, the dormouse weighed 17 g and showed emaciated body condition (Figure 2). The cardia to the body of the stomach was occupied by a 0.8 cm in diameter, locally extensive, tan-colored, raised but not projecting foci (Figure 3). Multiple ulcerative lesions were observed in the thickened region. The lungs were mottled and red. The clinician claimed that neither skin mass nor other gross lesions of internal organs were observed. Organs including heart, liver, spleen, lungs, kidneys, adrenal glands, stomach, and several segments of the intestine were collected and submitted to Animal Disease Diagnostic Center, National Chung Hsing University for pathological examination. The cross-section of the fixed stomach tissue revealed three or four folds (3–4 mm) of thickened, tan-colored, firm mucosal layer (Figure 4).

Microscopically, the mucosa to the submucosa of the thickened foci expanded by an unencapsulated, infiltrative neoplasm (Figure 5). Locally extensive gastric ulceration with chronic active inflammation was noted in the central area of the neoplasm. The neoplasm was composed of sheets or cord patterns of round cells which were interlaced with granulation tissue and moderate numbers of lymphocytes, plasma cells, and macrophages (Figure 6). The borders of neoplastic cells were indistinct. Most neoplastic cells were round to oval and occasionally elongated. The nucleus: cytoplasm ratio was 1:1 to 1:4, with moderate to abundant amount of amphophilic, smudged cytoplasmic granules. The nuclei of tumor cells were round to oval, centrally located with coarsely stippled nucleoli. Mitotic figures were rarely observed (<1 per 10 high power fields). There were moderate numbers of eosinophils infiltrated between neoplastic cells. The submucosa beneath the neoplasm was thickened and separates the mucosa muscularis and the tunica muscularis. There were moderate numbers of neoplastic cells, eosinophils, lymphocytes, and plasma cells surrounding the vessels. Toluidine blue stain showed >75% of neoplastic cells within the lamina propria and submucosa with strong metachromasia while the lymphocytes, eosinophils, and granulation tissue remained negative (Figure 7). The immunohistochemical staining (IHC) profile is listed in Table 1. Normal tissues from the same dormouse and a hamster served as the internal and external controls, respectively. Negative controls consisted of omission of the primary antibody. IHC for KIT reveals strong membranous pattern in neoplastic cells (Figure 8). IHC for other biomarkers including CK, chromogranin A, and CD79a revealed no immunoreactivity in the neoplastic cells. No metastasis was observed in sections of other organs. The final diagnosis of gastrointestinal mast cell tumor was made based on the histopathological, histochemical, and immunohistochemical characteristics.

Microscopic lesions of other organs including bronchus-associated lymphoid tissue hyperplasia, pulmonary hemorrhage, multifocal myocardial fibrosis with lipofuscin, and extramedullary hematopoiesis in the spleen and liver were noted.

## 3. Discussion

In the current case, the alimentary signs, including vomiting and anorexia, correspond to signs in gastrointestinal MCTs (GIMCTs) in other species [4,6,18,19]. Focal, hypoechoic wall thickening of the affected digestive tract is also the most common sonographic finding in GIMCTs in cats [20]. However, other gastrointestinal inflammation or neoplasia may also contribute to similar clinical findings. Based on the pathological findings of infiltration of round cells with amphophilic cytoplasmic granules in the gastric mucosa, differential diagnoses should include mast cell tumor, large granular lymphocytic lymphoma, carcinomas, and neuroendocrine tumors [6]. Metachromatic cytoplasmic granules under toluidine blue stain in combination with positive immunostaining for KIT have been suggested to be a reliable tool for the diagnosis of GIMCTs [3,6,21]. The diagnosis of GIMCT in this dormouse was confirmed with colocalization of positive, membranous immunostaining for KIT with strong and abundant metachromatic cytoplasmic granules in most tumor cells. Other tumors, including carcinomas, neuroendocrine tumors, and B cell lymphoma were excluded according to the result of the immunohistochemistry (IHC) panel. However, T cell lymphoma and histiocytic neoplasms should be taken into consideration since antibodies against these cell types could not perform adequately in this case (data not shown). The clinical signs and lesions indicate that the GIMCT had heavily affected the integrity of the gastric mucosa and led to the death of the dormouse. Mucosal ulceration is usually associated with direct tumor cell invasion in GIMCT, though excessive release of histamine through degranulation of neoplastic mast cells may also contribute to the digestive signs [5,6,19]. Extramedullary hemopoiesis in the liver and spleen also support the identification of long-term hemorrhage, which is thought to be caused by gastric ulceration.

In cats, the most GIMCTs-affected species, the GIMCTs are typically located in the submucosa and the tunica muscularis or transmural expansion while mucosal infiltration is usually minimal [6]. In contrast, GIMCTs in dogs often reveal diffuse tumor cell invasion from the mucosal cell layer to the tunica muscularis or serosa, sometimes replacing all layers of the gastrointestinal wall [19]. In the current case, gastric lamina propria is the most affected layer while the submucosa shows moderate infiltration. Since GIMCT has not been reported in Rodentia, we suggest that the phenotype of this dormouse is more similar to GIMCTs in dogs than those in cats. The difference in microscopic location of GIMCTs may associate with the origin of the tumor cells. Mast cells can be divided into two subtypes according to their locations: connective tissue mast cells (CTMCs) or mucosal mast cells (MMCs) [19,22]. Because of the difference in components of cytoplasmic granules, CTMCs can be distinguished from MMCs by metachromasia under toluidine blue stain, and immunohistochemical stain for tryptase, chymase, heparin, and chondroitin sulfate [3]. MMCs are thought to be non-metachromatic in several domestic species [3,19,22]. Since these IHC markers have not been performed in this case, we cannot determine whether the tumor origin was CTMCs or MMCs. Unlike granules of mast cells in other domestic animal species and humans, metachromasia is prominent in this dormouse regardless of the subtypes. However, we highly suggest that the origin of this case belongs to the mucosal mast cells population on the basis of the predominant mucosal infiltration of the tumor cells. Further elucidation of the components of cytoplasmic granules, their biological functions, and the standard methods to distinguish CTMCs and MMCs in *Graphiurus* spp. is needed.

The *c-KIT* gene encodes KIT, a transmembrane tyrosine kinase receptor protein that is involved in the development, proliferation, and function of mast cells and other cell populations [2,21,23]. In humans, dogs, and cats, the KIT mutations often occur in the neoplastic mast cells and result in altered KIT protein structure and function [2,21,23,24]. In dogs, certain *c-KIT* mutations are associated with aberrant protein localization (cytoplasmic pattern) and more malignant and lethal disease [21]. In contrast, *c-KIT* mutations in cat and human mast cell neoplasms are not correlated with prognosis [2]. *c-KIT* genetic abnormalities have not been well investigated in other species. The tumor cells in gastric mucosa, CTMCs, and MMCs of the current case all show membranous pattern which is thought to be normal. The tumor cells in this dormouse also share similar appearance and stain characteristics with normal mast cells except as to the relatively wide range of cell size. Further exploration of the association between the *c-KIT* pattern and the malignancy of GIMCTs in dormice is needed.

Herein, we report the first case report of MCT in dormouse but also the first GIMCT in rodent species. The diagnosis of GIMCT should be taken into consideration in dormice and other small rodent species with chronic vomit or hematemesis and should be treated with caution because of the disparity between the cell differentiation and clinical outcome.

## Figures and Tables

**Figure 1 vetsci-09-00497-f001:**
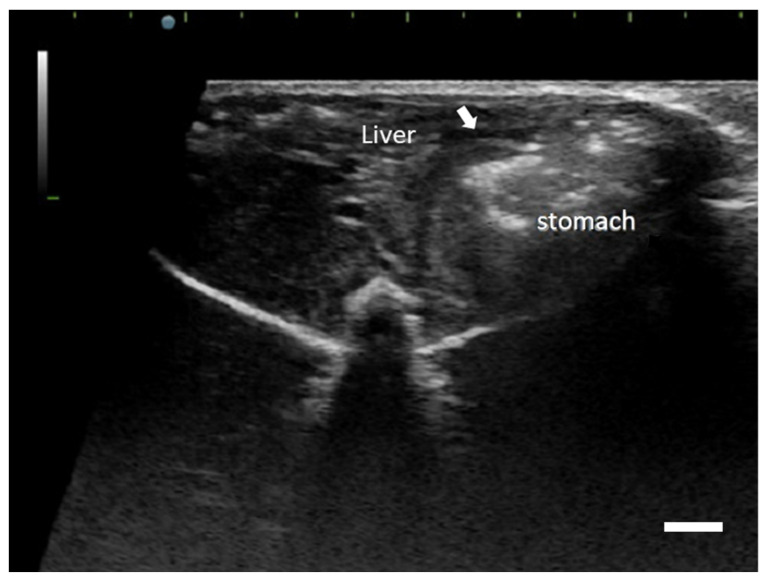
Gastric mast cell tumor, African dormouse. Transverse abdominal ultrasound image reveals focal, hypoechoic thickened foci of the gastric mucosa (arrow). Scale bar = 0.2 cm.

**Figure 2 vetsci-09-00497-f002:**
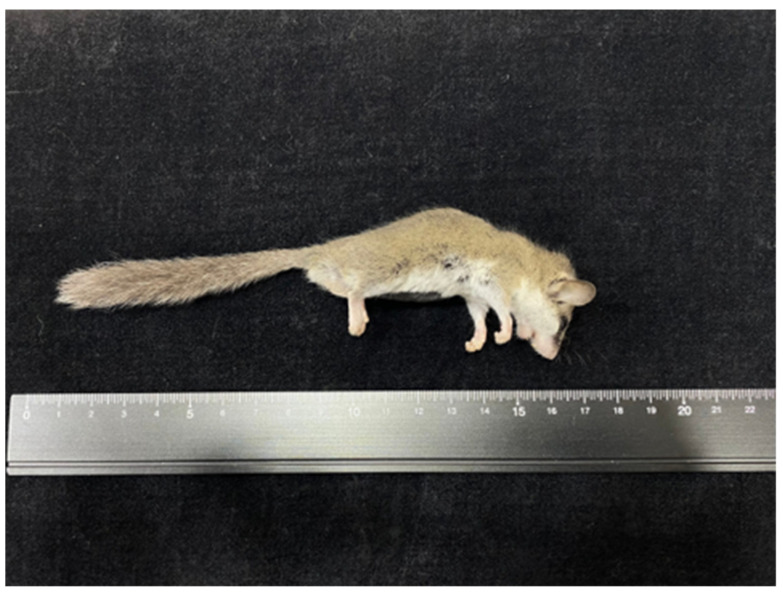
Gastric mast cell tumor, African dormouse. The dormouse is emaciated with abdominal retraction.

**Figure 3 vetsci-09-00497-f003:**
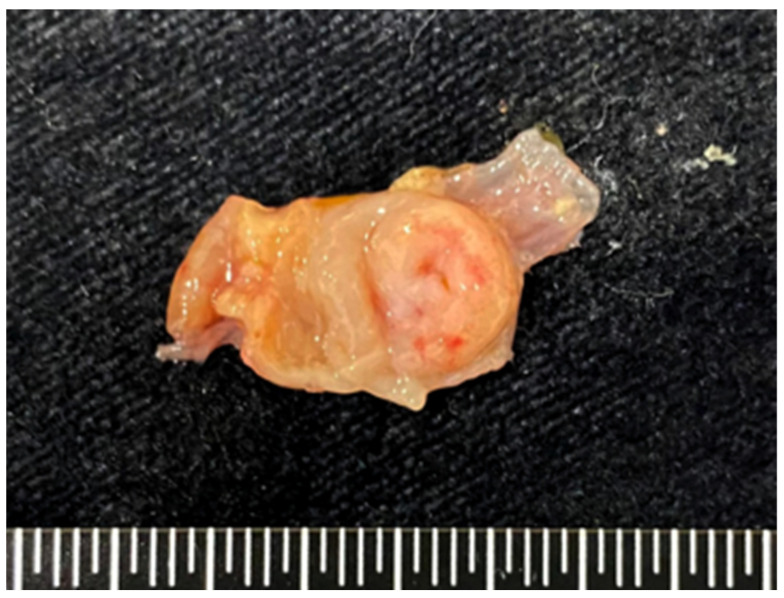
Gastric mast cell tumor, African dormouse, Stomach. A locally extensive, pale, thickened mucosal foci occupy the cardia to the body of the stomach.

**Figure 4 vetsci-09-00497-f004:**
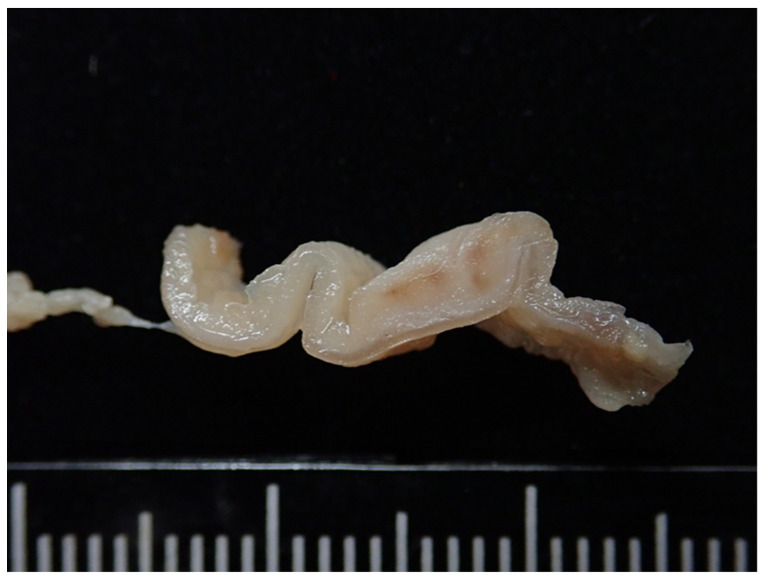
Gastric mast cell tumor, African dormouse, Stomach. The cross-section of the fixed stomach tissue revealed thickened tan-colored, firm mucosal layer.

**Figure 5 vetsci-09-00497-f005:**
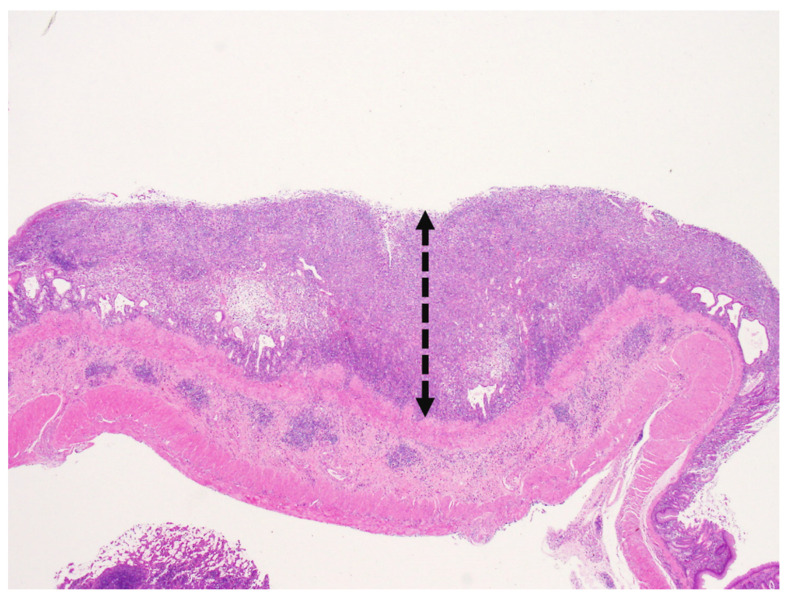
Gastric mast cell tumor, African dormouse, Hematoxylin and eosin. Focally and markedly thickened mucosa is expanded by an unencapsulated, infiltrative neoplasm with ulceration and dilation of the gastric glands (20×).

**Figure 6 vetsci-09-00497-f006:**
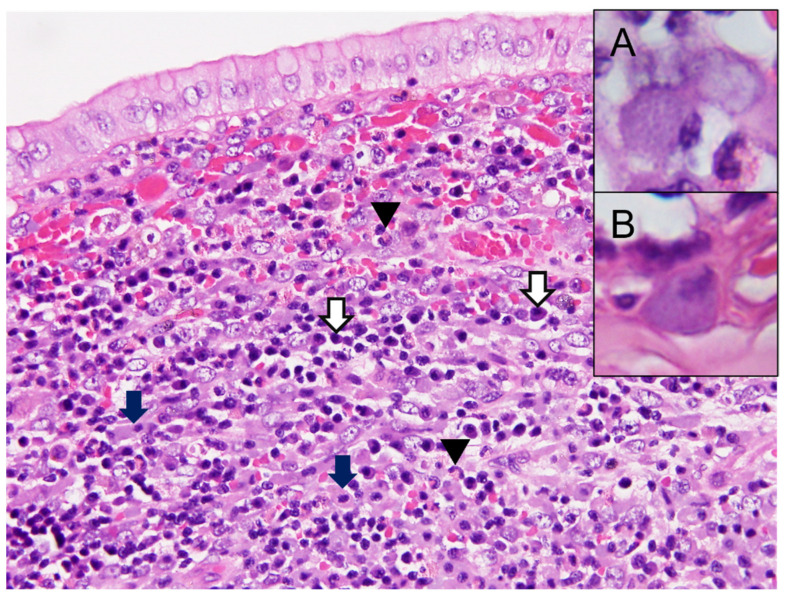
Gastric mast cell tumor, African dormouse, Hematoxylin and eosin. In the lamina propria, the tumor cells in sheet or cord patterns are mildly polymorphic (arrows) and are admixed with moderate number of eosinophils (arrowheads) and plasma cells (hollow arrows) (400×). There are amphophilic granules within the cytoplasm of neoplastic cells (inset **A**) (1000×). These features can also be observed in normal mucosal mast cells (MMCs) in the small intestine (inset **B**) (1000×).

**Figure 7 vetsci-09-00497-f007:**
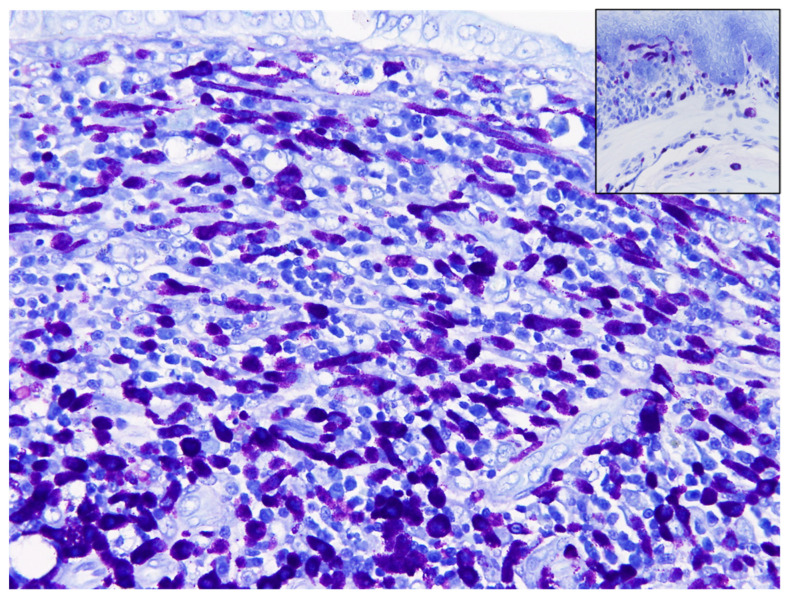
Gastric mast cell tumor (MCT), African dormouse, stomach, Toluidine blue stain. The neoplastic cells show strong metachromasia (400×). Both of MMCs and connective tissue mast cells (CTMCs) in the esophagus also show metachromasia (inset) (400×).

**Figure 8 vetsci-09-00497-f008:**
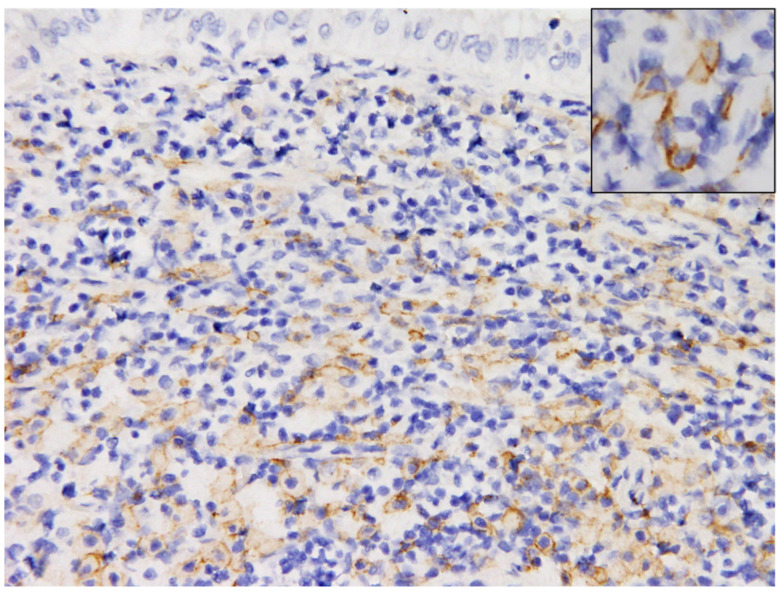
Gastric mast cell tumor (MCT), African dormouse, Immunohistochemistry against KIT. Some of the neoplastic cells reveal strong membranous pattern (400×). The same pattern can also be observed in the normal MMCs (inset) (400×).

**Table 1 vetsci-09-00497-t001:** Antibodies used for the immunohistochemical evaluation in this case.

Primary Antibody	Source	Clone	Dilution	Antigen Retrieval	Results ^1^
Rabbit anti-human KIT	Genemed	Polyclonal	1:100	EDTA	+
Mouse anti-human CK	Leica	AE1/AE3	1:800	enzyme	-
Rabbit anti-human Chromogranin A	Genemed	Polyclonal	1:100	EDTA	-
Mouse anti-human CD79a	Leica	JCB117	1:100	EDTA	-

^1^ +: Positive; -: Negative.

## Data Availability

Data are contained within the article.

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
