# Peer review of "Gastrointestinal Mast Cell Tumor in an African Dormouse (Graphiurus sp.)"

_vetsci, 2022, doi:10.3390/vetsci9090497_

Round 1

Reviewer 1 Report

The authors present a thorough clinical and postmortem work up of a rodent with gastric round cell neoplasm, confirmed as mast cell origin via  histochemical and immunohistochemical methods. The text is overall well-written though some stylistic changes are recommended below. The histologic images are very nice. The gross images could potentially be cropped a little more to highlight the lesions. The cytologic image should be re-taken as the resolution and lighting is not optimal. Generally, this will be of interest to clinicians and pathologists working with rodent species.

Example:

Lines 76-85: This paragraph starts as past tense then changes to present tense.

Reviewer 2 Report

The article is well-written (except by minor errors), well-organized and contains good development with the information presented in each section. In my knowledge and reviewing the current literature, I agree with the authors that this manuscript reports the first case of MCT in dormouse and the first GIMCT in rodent species.

There are few comments and questions that I am mentioning below:

Text

1.       Line 15. Muscularis is misspelled (Mascularis)

2.        Line 45 has reference 16 related to the introduced the monkeypox virus into the U.S. However, this paper is about “Experimental infection of an African dormouse (Graphiurus kelleni) with monkeypox virus”. I suggest the authors to include the original study about this investigation, which is available online (Reference: Centers for Disease Control and Prevention (CDC). Update: multistate outbreak of monkeypox--Illinois, Indiana, Kansas, Missouri, Ohio, and Wisconsin, 2003. MMWR Morb Mortal Wkly Rep. 2003 Jul 4;52(26):616-8. PMID: 12844080.

3.       The line 55, the word “dysemesis” was used. Not sure if this word exists and what is it meaning. Please double check it and maybe use a synonymous word that is more widely known.

4.       Line 56 (After receiving medication including Trimethoprim-Sulfamethoxazole (20mg/kg 56 BW), metoclopramide (1mg/kg BW), mosapride (0.5mg/kg BW), famotidine (1mg /kg BW) 57 twice a day for 1 week. ).  Need to review this phrase, seems that is missing a conclusion. After receiving ... what?

5.       Line 66 (At necropsy, the dormouse showed emaciated body condition). I recommend the authors to include the animal’s weight.

6.       Line 67 (A locally extensive, pale, thickened mucosal foci occupying the cardia to the body of the stomach was noted). I recommend the authors to include more detail about the gross aspect. Was the mass projecting into the lumen? Size? Extension of the thickened mucosa?

7.       Line 70 (observed. Vital organs…). I suggest excluding “vital”.

8.       Line 83,84 (moderate to abundant amount of amphophilic, smudged cytoplasm composed of varied-83 size granules) Is the majority of the tumor composed by a poorly granulated cells?

9.       Line 89. I am suggesting rewording this phrase "Toluidine blue stain showed >75% of neoplastic cells within the lamina propria and submucosa with strong metachromasia while the lymphocytes, eosinophils, and granulation tissue remain negative”

10.    Line 100. Please include in which tissue the extramedullary hematopoiesis was observed.

11.     Line 117 (Table 1). Question: Was the antibodies used independently validated in your laboratory using positive and negative controls? It is important to mention that. Also, in my opinion there is no contribution to list here, the antibodies that did not work. If the internal control did work, the IHC result should be inadequate. Indeterminate result is normally used when the immunostaining is rare/scarce or has inappropriate distribution and/or localization for target antigen(s).

12.    Line 118. Negative is misspelled (Negaive)

13.     Line 131-133 (Other tumors including carcinomas, neuroendocrine tumors, T cell lymphoma were excluded according 132 to the result of the immunohistochemistry (IHC) panel) This phrase might need to be reviewed if the IHC with inadequate (indeterminate) results are removed. Specially related with the DD for T cell lymphoma (this cannot be excluded based on the IHC for Cd3, since the IHC for CD3 is inadequate)

Figures

1.     Include magnification in all microscopic descriptions (is always better).

2.     Include arrows in the pictures 5 and 6.

3.     Figure 1. Use arrows for identification of the cells. I cannot say for sure if that are epithelial or mast cells.

4.     Figure 6. I do not see granules; maybe say that these cells are poorly granulated

5.     The legend of the Figure 8 says that “The neoplastic cells reveal…”. Change to " Some of the neoplasic cells...", since the immunostaining is not in all the cells.

Reviewer 3 Report

In the current case report, authors presented an African dormouse with gastrointestinal mast cell tumor. The strength point of this case is that it is rare to occur and to be diagnosed. The authors tried to stress this point and presented significant recommendations in this regard. However, to enhance the quality of the article, the following points are important.

1. In the abstract (Line 11), Authors mentioned that some findings of the sonography. More details should be mentioned in the case presentation.

2. Lines 62-63, authors mentioned some findings of whole body radiography. To make the case presentation more comprehensive and since this report describes that case for the first time, it is recommended to add some of the sonographic and radiographic images that describe your findings.

3. The English language should be improved.

Round 2

Reviewer 3 Report

The authors have carefully and extensively revised the manuscript. All  my comments have been addressed. I do not have any significant comment to add. I believe that the quality of the work presentation has been improved. Hence, I recommend publication of this case report in the present form.